# Mean Oral Cavity Organ-at-Risk Dose Predicts Opioid Use and Hospitalization during Radiotherapy for Patients with Head and Neck Tumors

**DOI:** 10.3390/cancers16020349

**Published:** 2024-01-13

**Authors:** Robert L. Foote, W. Scott Harmsen, Adam C. Amundson, Alan B. Carr, Mauricio E. Gamez, Yolanda I. Garces, Scott C. Lester, Daniel J. Ma, Lisa A. McGee, Eric J. Moore, Michelle A. Neben Wittich, Samir H. Patel, David M. Routman, Jean-Claude M. Rwigema, Kathryn M. Van Abel, Linda X. Yin, Olivia M. Muller, Satomi Shiraishi

**Affiliations:** 1Department of Radiation Oncology, Mayo Clinic, 200 First St. SW, Rochester, MN 55905, USA; amundson.adam1@mayo.edu (A.C.A.); gamezharo.mauricio@mayo.edu (M.E.G.); garces.yolanda@mayo.edu (Y.I.G.); lester.scott@mayo.edu (S.C.L.); ma.daniel@mayo.edu (D.J.M.); neben.michelle@mayo.edu (M.A.N.W.); routman.david@mayo.edu (D.M.R.); 2Department of Quantitative Health Sciences, Mayo Clinic, 200 First St. SW, Rochester, MN 55905, USA; harmsen.william@mayo.edu; 3Department of Dental Specialties, Division of Esthetic and Prosthetic Dentistry, Department of Advanced Prosthodontics, Mayo Clinic, 200 First St. SW, Rochester, MN 55905, USA; carr.alan@mayo.edu (A.B.C.); muller.olivia@mayo.edu (O.M.M.); 4Department of Radiation Oncology, Mayo Clinic Arizona, 5777 E. Mayo Blvd., Phoenix, AZ 85054, USA; mcgee.lisa@mayo.edu (L.A.M.); patel.samir@mayo.edu (S.H.P.); rwigema.jean@mayo.edu (J.-C.M.R.); 5Department of Otolaryngology-Head and Neck Surgery, Mayo Clinic, 200 First St. SW, Rochester, MN 55905, USA; moore.eric@mayo.edu (E.J.M.); vanabel.kathryn@mayo.edu (K.M.V.A.); yin.linda@mayo.edu (L.X.Y.); 6Division of Medical Physics, Mayo Clinic, 200 First St. SW, Rochester, MN 55905, USA; shiraishi.satomi@mayo.edu

**Keywords:** radiotherapy, head and neck cancer, oral cavity, organ at risk, adverse events

## Abstract

**Simple Summary:**

Approximately 75% of all head and neck cancer patients are treated with radiotherapy. Radiotherapy to the oral cavity results in acute and late adverse events which can be severe depending on the outcome measured and intensity of the treatment. These adverse events are detrimental to the patient’s quality of life and function and are costly to manage. The aim of this retrospective study was to evaluate the association between radiotherapy dose to a defined oral cavity organ-at-risk avoidance structure and prospectively obtained validated provider- and patient-reported outcomes, opioid use, and hospitalization in 196 patients treated with radiotherapy for tumors in the head and neck region. We identified a statistically significant association between radiotherapy dose to the oral cavity organ-at-risk structure, opioid use, and hospitalization. This association provides valuable information for radiotherapy treatment planning to allow for reduction in adverse events.

**Abstract:**

Background: Approximately 75% of all head and neck cancer patients are treated with radiotherapy (RT). RT to the oral cavity results in acute and late adverse events which can be severe and detrimental to a patient’s quality of life and function. The purpose of this study was to explore associations between RT dose to a defined oral cavity organ-at-risk (OAR) avoidance structure, provider- and patient-reported outcomes (PROs), opioid use, and hospitalization. Methods: This was a retrospective analysis of prospectively obtained outcomes using multivariable modeling. The study included 196 patients treated with RT involving the oral cavity for a head and neck tumor. A defined oral cavity OAR avoidance structure was used in all patients for RT treatment planning. Validated PROs were collected prospectively. Opioid use and hospitalization were abstracted electronically from medical records. Results: Multivariable modeling revealed the mean dose to the oral cavity OAR was significantly associated with opioid use (*p* = 0.0082) and hospitalization (*p* = 0.0356) during and within 30 days of completing RT. Conclusions: The findings of this study may be valuable in RT treatment planning for patients with tumors of the head and neck region to reduce the need for opioid use and hospitalization during treatment.

## 1. Introduction

There are over 900,000 cases of head and neck cancer diagnosed worldwide each year [1]. Approximately 75% of all head and neck cancer patients are treated with radiotherapy (RT) primarily or in combination with surgery and/or chemotherapy [2]. The oral cavity submucosa contains minor salivary glands which provide lubrication and moisture for the oral cavity [3]. The oral cavity mucosa contains taste buds which are necessary for the sensation of taste [4]. Radiotherapy to the oral cavity is associated with acute adverse events including painful oral mucositis, dysgeusia, sticky salivary secretions, dysphagia, and odynophagia [5,6,7,8,9,10,11,12,13,14]. These adverse events result in secondary use of opioid pain medications, dehydration, weight loss, malnutrition, use of intravenous fluids, feeding tubes, medications to thin salivary secretions, and hospitalization [5,6,7,8,9,10,11,12,13,14]. Acute toxicity can be severe or worse in 1–81% of patients depending on the outcome measured and intensity of the treatment [5,6,7,8,9,10,11,12,13,14]. Late adverse events include opioid pain medication dependency, mucosal infections, dysphagia, dysgeusia, altered diet with poor nutrition, food poisoning, sticky secretions, xerostomia, dental caries and extractions, osteoradionecrosis, stress, anxiety, anorexia, and depression [5,6,9,11,14,15,16]. Late toxicity can be severe or worse in 1–54% of patients depending on the outcome measured and intensity of the treatment [5,6,9,11,14,15,16]. These adverse events are detrimental to the patient’s quality of life and function and are costly to manage [5,6,9,11,14,15,16,17,18,19,20,21,22,23].

A variety of oral cavity organ-at-risk (OAR) avoidance structures have been defined to aid in RT treatment planning (File S1, [24,25,26,27,28,29]). The treatment goal is to limit the volume of the oral cavity OAR avoidance structure exposed to any RT dose to reduce the incidence and severity of acute and late adverse events. Currently there is no consensus on how to define the oral cavity OAR avoidance structure (File S1). Likewise, there is limited data available describing the association between the RT dose, the volume of oral cavity exposed to RT, and the incidence and severity of acute and late adverse events [24,27,30,31,32,33,34,35,36,37,38,39,40,41,42,43,44,45,46,47,48,49,50,51,52,53,54]. The goal of this study was to define an oral cavity OAR avoidance structure and to determine if there is a dose and volume relationship for the oral cavity OAR avoidance structure as defined which would be predictive of acute and late adverse events, and which would be useful for RT treatment planning. We found that the mean dose to the defined oral cavity OAR avoidance structure was significantly associated with opioid use and hospitalization during and within 30 days of completing RT.

## 2. Materials and Methods

This retrospective review was approved by the Mayo Clinic Institutional Review Board (IRB #20-004915, 17 June 2020). Inclusion criteria included all patients with benign or malignant tumors within the head and neck region receiving RT to the oral cavity administered by one of the authors (RLF) between 1 January 2017 and 31 December 2020. The Mayo Clinic Rochester Department of Radiation Oncology Patient Outcomes Database (IRB #15-00136, 15 March 2015) was searched electronically by provider (RLF), date (1 January 2017 to 31 December 2020), and International Classification of Diseases (ICD)-10 codes to identify eligible patients. Appendix A provides the ICD-10 codes included in the search. Exclusion criteria included a palliative course of hypofractionated RT, prior RT to the oral cavity, no oral cavity OAR within the RT treatment volume (no dose to the oral cavity), hypofractionated RT for melanoma, hypofractionated stereotactic body RT, and no consent provided to use medical records for research purposes as provided either by the Department of Radiation Oncology Registry (IRB #15-000136, 15 March 2015) or by Minnesota state statute. Figure 1 is a Consolidated Standards of Reporting Trials (CONSORT) diagram for patient inclusion and exclusion.

Presence of diabetes mellitus, hypertension, and leukoplakia prior to RT, use of concurrent chemotherapy, smoking status at the time of treatment, use of percutaneous endoscopic gastrostomy (PEG) tube, and weights were collected using a combination of chart review and data extraction from Mayo Clinic’s Unified Data Platform (UDP) which is a data warehouse for current and historical medical records. For diabetes mellitus, hypertension, and leukoplakia, diagnosis codes were retrieved from medical records. If the diagnosis date was before the RT end date, then the patient was considered to have had a co-morbidity during RT. Whether a patient received concurrent chemotherapy was collected from treatment summary notes. The medication administration database for specific drugs was searched to identify if the chemotherapy was cytotoxic therapy, hormonal therapy, or immunotherapy, and to identify patients prescribed opioid pain medications 30 days prior to the start of RT, during RT, and up to 30 days after the completion of RT. Smoking status closest to the RT start date was extracted, and patients were placed into one of two groups: never smokers and former or current smokers. PEG tube insertion procedure codes were searched to identify patients who utilized a PEG tube after the onset of RT (2 patients with PEG tubes were excluded because the PEG tube placement was unrelated to the malignancy or the treatment). Patient weights were also collected from the UDP and evaluated at baseline prior to RT, post RT, and 3 months, 6 months, 12 months, and 24 months following the completion of RT. Glossectomy status was determined by ICD-10 code (C01 or C02) and treatment (postoperative RT). Hospitalization during and up to 30 days after the completion of RT was recorded from the Department of Radiation Oncology Hospitalization Dashboard.

Radiotherapy treatment plans were created, and dose volume histograms (DVH) were generated using the Eclipse treatment planning system (Varian Medical Systems, Palo Alto, CA, USA, www.varian.com, accessed on 2 January 2024). The standard photon treatment plans consisted of three volumetric modulated arc therapy fields. The standard proton treatment plans consisted of three to four fields with the pencil beam scanning technique. For proton therapy plans, the RT dose was scaled by 1.1 to account for the difference in relative biological effect when compared to conventional photon treatments. A definition for the oral cavity OAR avoidance structure evolved at our institution between 2003 and 2016 when it was formalized and accepted as the consensus standard for treatment planning (Figure 2A–H). The DVH statistics for the oral cavity OAR and the mean dose to the left and right submandibular glands, left and right parotid glands, and total parotid glands were calculated and extracted. Pharyngeal constrictor, intrinsic and extrinsic muscles of the tongue, and laryngeal DVH statistics were not included since swallowing function is not a primary endpoint for this study. There were four patients who did not complete the planned course of treatment. In these cases, the prescribed dose was scaled to the delivered dose.

The superior extent of the oral cavity OAR avoidance structure includes all the mucosa of the hard palate. The contouring of the OAR begins superiorly at the first sign of mucosa on the alveolar ridge of the maxilla (medial and lateral) and hard palate. It then continues inferiorly to include the mucosa of the upper and lower lip, mucosa of the hard and soft palate including the uvula, the buccal mucosa including the buccinator muscles, the mucosa of the retromolar trigone, the entire tongue (anterior two-thirds, dorsal surface, and tongue base), floor of mouth, sublingual glands, gingival mucosa of the mandible (lingual and buccal surfaces), and ends at the level of the cranial edge of the hyoid bone and caudal edge of the mandible. It also includes the maxillary and mandibular teeth if present. The posterior extent includes the soft palate, uvula, and tongue base. The anterior extent includes the mucosal surface of the posterior one-half of the lips and the gingival mucosa of the maxillary and mandibular alveolar ridges and retromolar trigone. The lateral extent includes the buccal mucosa and buccinator muscles. The oral cavity OAR contains most of the taste buds, which are located within the mucosa of the anterior two-thirds of the tongue, the floor of the mouth, the buccal mucosa, the lips, the pharynx (including the soft palate, uvula, and base of the tongue), the larynx (epiglottis), and the upper third of the esophagus [4]. The oral cavity OAR also contains the minor salivary glands located within the buccal, labial, lingual, soft palate, lateral parts of the hard palate, and floor of the mouth submucosa and in the trough circling the circumvallate papillae on the dorsal surface of the tongue near the terminal sulcus [3]. Therefore, the oral cavity OAR for radiotherapy (RT) treatment planning purposes is defined as including the anterior two-thirds of the tongue, floor of mouth, buccal mucosa, mucosal surface of the lips, soft palate, uvula, base of tongue, hard palate, and circumvallate papillae on the dorsal surface of the tongue. The gingival mucosa of the alveolar ridges of the mandible and maxilla and the mucosa of the retromolar trigone are also included in the definition to further reduce the incidence and severity of painful oral mucositis. Finally, the sublingual glands are included in the oral cavity OAR structure. The larynx (epiglottis) and upper third of the esophagus are not included in this OAR volume because they are included in their own OAR avoidance structure (larynx, cricopharyngeal inlet, cervical esophagus).

Provider-reported adverse events, using the Common Terminology Criteria for Adverse Events (CTCAE) Version 4.03 (Published: 14 June 2010. US Department of Health and Human Services, National Institutes of Health, National Cancer Institute), and patient-reported outcomes, using the European Organisation For Research and Treatment Of Cancer Quality of Life Questionnaire Head and Neck (EORTC QLQ-H&N35) and the Patient-Reported Outcomes Measurement Information System (PROMIS) Global-10, were collected prospectively at baseline prior to RT, at completion of RT, and at 3 months, 6 months, 12 months, and annually thereafter following completion of RT [55,56]. Hospitalization and the start of opioid pain medications during and up to 30 days after the completion of RT were also recorded. Patients prescribed opioid pain medications within 30 days of starting RT were presumed to be taking opioid pain medications at the start of RT and were excluded from the analysis of this endpoint. Patients were censored at last follow-up if alive, at death, or when a second course of RT was administered to the oral cavity. The primary endpoints of interest were the incidence of CTCAE v4.03 ≥ grade 2 oral pain, salivary duct inflammation, dysgeusia, dehydration, and dry mouth; EORTC QLQ-H&N35 subscales for senses, saliva, speech, and social eating with ≥10 point change, and scores for pain, pain medication, feeding tube use, and weight loss; PROMIS Global-10 physical and mental health subscale T-scores with ≥10 point change, measured weight loss ≥10% at end of treatment, 3 months, 6 months, 12 months, and 24 months following completion of RT; start of opioid pain medication use during and up to 30 days after the completion of RT; and hospitalization during and up to 30 days after the completion of RT. Secondary endpoints of interest were CTCAE v4.03 ≥ grade 2 dysphagia and EORTC QLQ-H&N35 subscales for swallow with ≥10-point change.

Patients presenting with baseline pre-RT CTCAE v4.03 adverse events secondary to surgery, malignancy, chemotherapy, medications, or co-morbid illnesses were excluded from the specific adverse event analysis. This included 2 patients with baseline CTCAE v4.03 ≥ grade 1 dehydration, 3 patients with ≥grade 2 dry mouth, 5 patients with ≥grade 2 dysgeusia, 11 patients with ≥grade 2 dysphagia, 5 patients with ≥grade 2 oral pain, and 4 patients with ≥grade 2 salivary inflammation.

CTCAE v4.03 adverse events at specified time points with insufficient ≥grade 2 events to run logistical models predicting ≥grade 2 toxicity were excluded from univariate logistic modeling. A decrease in the PROMIS Global-10 physical health T-score or mental health T-score of ≥10 points is considered clinically significant [57]. A decrease in the EORTC QLQ-H&N35 subscale score ≥10 points is considered clinically significant [58]. Variables included in the univariate analysis included age, sex, smoking status, diabetes mellitus, hypertension, cytotoxic chemotherapy, oral cavity OAR D_max_, mean dose, and V10Gy-V70Gy in 10 Gy increments, mean right, left, and total parotid gland dose, mean right and left submandibular gland dose, RT modality, and glossectomy. For the multivariable logistic models, 14 outcomes having at least 30 events (or at least 30 non-events if the number of non-events is fewer than events) and having a univariate *p*-value of <0.05 were included in full and parsimonious multivariable modeling.

The data analysis plan was to evaluate DVH statistics for the oral cavity OAR avoidance structure (D_max_ [maximum dose to 0.01 cc], mean dose, and V10Gy to V70Gy in 10 Gy increments where VxGy is the volume of oral cavity OAR receiving xGy or more) to determine if there is a statistically significant association with provider-reported adverse events (CTCAE v4.03), patient-reported outcomes (EORTC QLQ-H&N35 subscales and PROMIS Global-10 physical and mental health T-scores), start of opioid pain medications during or within 30 days of completing RT, and/or hospitalization during or within 30 days of completing RT at specified times.

Logistic regression models were used to identify predictive clinical parameters for the endpoints of this study. Endpoints included provider-reported adverse events using CTCAE v4.03 > grade 2 oral pain, salivary duct inflammation, dysgeusia, dehydration, dysphagia, and dry mouth; patient-reported outcomes using EORTC QLQ-H&N35 subscales for swallow, senses, speech, social eating, and saliva with >10 point change, and scores for pain, pain medication, feeding tube use, and weight loss; PROMIS Global-10 T-scores for physical and mental health with ≥10 point change; measured weight loss at end of treatment, 3 months, 6 months, 12 months, and 24 months following completion of RT; starting opioid pain medication during and up to 30 days after the completion of RT; and hospitalization during and up to 30 days after completion of RT. A multivariable logistic model was identified for each outcome including those predictors deemed clinically relevant. A parsimonious model was identified retaining those variables with statistical significance of <0.05. The alpha level was set at 0.05 for statistical significance. All analyses were completed using SAS version 9.4 (SAS Institute Inc. 2019). Pearson correlations were used to evaluate the correlation between oral cavity OAR DVH statistics. DVH statistics were compared between RT modalities using a two-sample *t* test. For DVH parameters, the total oral cavity OAR avoidance structure volume was evaluated for correlation with provider-reported adverse events (CTCAE v4.03), patient-reported outcomes (EORTC QLQ-H&N35 subscales, PROMIS Global-10 subscales), opioid pain medication use, and hospitalization. Sensitivity and specificity estimates from a receiver operating characteristic curve were used to identify cut-points to achieve at least 90%, 80%, and 70% sensitivity and maximize the subsequent specificity.

## 3. Results

Table 1 summarizes the patient, tumor, and treatment characteristics for the 196 patients included. The study group included 142 males (72.4%). The median age was 63.0 years (IQR 55.1, 72.5).

### 3.1. Univariable Analysis

The univariable analysis revealed the oral cavity OAR DVH statistics were associated with the following CTCAE v4.03 adverse event endpoints: dehydration > grade 1 at end of treatment; dry mouth > grade 1 at end of treatment, 3 months, 6 months, 12 months, and 24 months; dysgeusia > grade 1 at end of treatment and 3 months; dysphagia > grade 1 at end of treatment, 3 months, 6 months, 12 months, and 24 months; oral pain > grade 1 at end of treatment; salivary duct inflammation > grade 1 at end of treatment and 3 months; and weight loss ≥ 10% at end of treatment, 3 months, 6 months, 12 months, and 24 months (see Appendix A for odds ratios and 95% Confidence Intervals).

The univariable analysis also revealed the oral cavity OAR DVH statistics were associated with a clinically significant decrease in the Promis-10 Physical T-score of ≥10 points at 3 months and a decrease in the Promis-10 Mental T-score of ≥10 points at 12 months (Appendix A).

The oral cavity OAR DVH statistics were also found to be associated with the following changes in EORTC QLQ-H&N35 subscales: swallow ≥ 10 points at 3 months, 6 months, and 24 months; saliva ≥ 10 points at end of treatment; senses ≥ 10 points at end of treatment and 6 months; speech ≥ 10 points at 24 months; social eating ≥ 10 points at end of treatment, 3 months, 6 months, and 12 months; and feeding tube use at end of treatment (Appendix A).

Finally, the oral cavity OAR DVH statistics were found to be associated with the start of opioid pain medications during and within 30 days of completing RT. The oral cavity OAR DVH statistics were also found to be associated with hospitalization during and within 30 days of completing RT (Appendix A).

Table 2 demonstrates the high correlation between mean oral cavity OAR dose and V10Gy-V70Gy. Therefore, only the mean oral cavity OAR dose was used in the multivariable analysis.

### 3.2. Multivariable Modeling

Table 3 summarizes the multivariable modeling. The mean oral cavity OAR dose was associated with the start of opioid pain medication during and within 30 days of completing RT (odds ratio [OR] 2, (95% Confidence Interval {CI} 1–2), *p* = 0.008) and with hospitalization during and within 30 days of completing RT (OR 1, 95% CI 1–2; *p* = 0.036). The model was driven largely by the dose to the major salivary glands. The use of cytotoxic chemotherapy was associated with a higher risk of developing CTCAE v4.03 dehydration >grade 1 at the end of treatment (OR 5, 95% CI 3–12; *p* < 0.001), the start of opioid pain medication during and withing 30 days of completing RT (OR 4, 95% CI 2–11; *p* = 0.002), and with hospitalization during and within 30 days of completing RT (OR 2, 95% CI 1–5; *p* = 0.022). The use of protons was associated with a lower risk of developing CTCAE v4.03 dysphagia > grade 1 at end of treatment and at 3 months (OR 0.3, 95% CI 0.1–0.6; *p* = 0.003 and OR 0.4, 95% CI 0.1–0.9; *p* = 0.029, respectively). The use of protons was also associated with a lower risk of developing CTCAE v4.03 salivary duct inflammation > grade 1 at end of treatment (OR 0.4, 95% CI 0.2–1; *p* = 0.047). The C-statistic suggests the models are good to strong.

Table 4 compares RT modality and oral cavity OAR DVH statistics. The dose to the oral cavity OAR was significantly lower when protons were used compared to photons.

The data was assessed to identify a mean oral cavity OAR dose cut point which could be used to guide RT treatment planning to minimize the risk of opioid pain medication use and hospitalization during and within 30 days of completing RT. Table 5 and Table 6 provide the mean oral cavity OAR dose cut points to achieve at least 90%, 80%, and 70% sensitivity for correctly predicting the need for opioid pain medications or hospitalization during or within 30 days of completing RT while maximizing specificity.

## 4. Discussion

A systematic review of acute taste impairment following RT suggested approximately 96% of patients will experience objective taste impairment, and approximately 79% will report subjective taste impairment [34]. In a prospective longitudinal study of taste impairment in patients with stage III and IV oropharyngeal carcinoma treated with intensity modulated radiotherapy (IMRT), severe taste impairment was reported by 50% of patients 1 month after completing treatment, 40% at 3 months, 22% at 6 months, and 23% at 12 months [24]. The University of Michigan definition of oral cavity OAR was used in the study. Taste impairment was measured using the Head-and-Neck QOL (HNQOL) questionnaire and University of Washington Head- and Neck-Related QOL (UWQOL) questionnaire. Patient-reported severe taste impairment was significantly associated with RT dose to the oral cavity and tongue. Normal tissue complication probability (NCTP) revealed the mean oral cavity dose at which 50% of patients develop severe taste impairment is 53 Gy and 57 Gy according to the HNQOL and UWQOL instrument results, respectively. The NCTP mean oral cavity dose at which 25% of patients develop severe taste impairment is 39 and 42 Gy, respectively. The present study did not reveal an association between oral cavity OAR dose and taste impairment on multivariable analysis using the CTCAE v4.03 and EORTC QLQ-H&N35 instruments. Possible explanations for this include the use of different definitions for the oral cavity OAR, the use of different patient-reported outcomes questionnaires, and different taste impairment severity endpoints (we used ≥grade 2 whereas the University of Michigan group used severe). We also excluded patients with pre-RT taste impairment. Finally, we performed a more extensive multivariable analysis whereas the University of Michigan group controlled for age, sex, and salivary flow rate.

Chen et al., prospectively evaluated patient-reported taste impairment using the EORTC QLQ-H&N35 taste-related question (question 44, 0 = no taste impairment, ≥33.3 = taste impairment) in patients undergoing curative primary or postoperative IMRT with or without concurrent chemotherapy [41]. They used the University of Michigan definition of oral cavity OAR [36]. They excluded patients treated with chemotherapy or RT for recurrent cancer or second primary tumors, patients using medications that could impair taste, and/or patients with occupational exposures that could cause impairment of taste. In multivariable analyses, partial or total glossectomy was significantly associated with long-term taste impairment. When the authors excluded patients treated with surgery from the analyses, the mean RT dose to the oral cavity was not associated with taste impairment. When the mean RT dose was <5000 cGy, 14.3% of patients experienced taste impairment. When the mean RT dose was ≥5000 cGy, 28.3% of patients experienced taste impairment. The authors concluded that glossectomy is a major cause of long-term taste impairment in head and neck cancer patients receiving IMRT. The study was limited by incomplete baseline data and a small sample size. In the present study, the multivariable analysis did not suggest that glossectomy had an impact on taste impairment. While both studies did come to the same conclusion, that there is no clinically significant relationship between the mean RT dose to the oral cavity OAR and subjective taste impairment using the EORTC QLQ-H&N35 questionnaire, it should be noted that different definitions for the oral cavity OAR were used and Chen et al. excluded patients who had a history of “use of drugs and/or occupational exposure to substances that could affect taste”. Such drugs and occupational exposures were not defined or delineated.

Fried et al. pooled patients with human-papillomavirus- or p16-related squamous cell carcinoma of the oropharynx (favorable risk) from three multi-institutional phase II studies evaluating deintensified treatments [31]. Exclusion criteria included less than 6 months of follow-up and patients without RT plans available for analysis. The authors do not state how many patients were excluded from their analysis. The oral cavity OAR was defined as the oral tongue, tongue base, floor of mouth, hard palate, soft palate, buccal mucosa, and lip mucosa (upper and lower). PRO-CTCAE related to dry mouth and taste impairment at 6 and 12 months were used as the outcomes for this study. Multivariable analyses revealed dry mouth severity at 6 months was significantly associated with the mean RT dose to the contralateral parotid gland, oral cavity OAR, and patient-reported dry mouth at baseline. Dry mouth severity at 12 months was significantly associated with baseline dry mouth and the mean RT dose to the contralateral submandibular gland. Taste impairment at 12 months was significantly associated with the mean RT dose to the oral cavity OAR. Evaluation of substructures within the oral cavity OAR revealed that dry mouth severity at 6 months was related to the mean dose to the floor of mouth. Taste impairment at 12 months was associated with mean dose to the oral tongue. The authors concluded the floor of mouth and oral tongue should be prioritized during RT treatment planning over the rest of the oral cavity OAR structures. The study had two limitations, (1) biases associated with pooled analyses and (2) missing patient-reported quality-of-life metrics data. In the present study, we did not find a statistically significant relationship between subjective dry mouth and the mean RT dose to the oral cavity OAR at 6 months. Similarly, we did not find a statistically significant relationship between subjective taste impairment and the mean RT dose to the oral cavity OAR at 12 months. Possible explanations for the differences in our findings include inclusion of different patient populations (the UNC group included only patients with HPV-related oropharynx or unknown primary cancers), the use of different patient-reported outcome questionnaires, and the use of different levels of severity endpoints (≥grade 2 vs. none/mild, moderate, and severe/very severe). We did not analyze the outcomes by substructures within the oral cavity OAR with the goal of avoiding adding additional burdensome contouring tasks of arbitrarily defined substructures. The use of machine learning/deep learning in the future may reduce the burden of substructure segmentation.

In the present study, we did not include patients with baseline xerostomia > grade 1. On multivariable analysis, mean dose to the parotid glands or oral cavity OAR was not associated with xerostomia using the provider-reported CTCAE v4.03. Xerostomia at end of treatment was associated with mean right submandibular gland dose. A clinically significant decrease in the EORTC QLQ-N&N35 saliva subscale of ≥10 points was also associated with the mean right submandibular gland dose. In addition, dysgeusia at 12 months using the provider-reported CTCAE v4.03 was not found to be related to mean dose to the oral cavity OAR. However, dysgeusia at end of treatment was related to mean dose to the right submandibular gland.

In the study by Chen et al., the whole-mouth solution method for four tastes (salt, sweet, sour, and bitter) was used to objectively measure taste function in patients with head and neck cancer treated with IMRT or volumetric modulated arc therapy (VMAT) [32]. Patients were excluded if they had received previous RT to head and neck regions and/or had abnormal taste function prior to RT. Additional endpoints included subjective provider evaluations using CTCAE v4.03 and the Subjective Total Taste Acuity scale. Patient self-reported quality of life was evaluated using EORTC QLQ-H&N35. The oral cavity OAR as defined by the University of Michigan was used [36]. The authors reported a positive correlation between the subjective perception of impaired taste and the objectively measured impairment of the four taste qualities. An oral cavity mean dose ≥4000 cGy was associated with objective taste impairment 3 months after completing RT. With a mean oral cavity RT dose < 4000 cGy, 15.6% of patients developed taste impairment. With a mean oral cavity dose ≥4000 cGy, 44.9% of patients developed taste impairment. The mean oral cavity NTCP doses at 3 months were 25 Gy (15%), 38 Gy (25%), and 60 Gy (50%). The mean oral cavity NTCP doses at 6 months were 57 Gy (15%), 60 Gy (25%), and 64 Gy (50%). The authors concluded that a high mean oral cavity dose is associated with objective taste impairment in patients with head and neck cancer receiving IMRT. Reducing oral cavity dose may promote early recovery of taste function after IMRT. Limitations of the study include data from a single institution with limited sample size. The present study, on multivariable analysis, did not reveal an association between taste impairment and mean dose to the oral cavity OAR as defined. Possible explanations for the differences reported in the present study and the above study include the use of different definitions of the oral cavity OAR and the use of different measurements for taste impairment, especially objective vs. subjective.

The univariable analysis reported in the present study revealed the oral cavity OAR DVH statistics were associated with the CTCAE v4.03 adverse event endpoints studied, a clinically significant decrease in the Promis-10 Physical T-score and Mental T-score, and the EORTC QLQ-H&N35 subscales studied (*p* < 0.05, Appendix A). However, on multivariable analysis, they were not.

The multivariable models revealed the CTCAE v4.03 adverse event endpoints of dehydration, dry mouth, dysgeusia, dysphagia, salivary duct inflammation, weight loss ≥ 10%, and EORTC QLQ-H&N35 salivary scale decrease ≥ 10 points were significantly associated with the use of cytotoxic chemotherapy, proton therapy, and mean dose to the parotid and submandibular glands. This highlights the impact of radiation dose to salivary glands on the important functions of saliva production, taste, and swallowing with subsequent weight loss.

The mean dose to the oral cavity OAR, as defined at our institution, was found to be significantly associated with the objective outcomes of the use of opioid pain medication and hospitalization during and within 3 months of completing RT. This would suggest the definition of the oral cavity OAR as used would be helpful in RT treatment planning to reduce the likelihood of these adverse events. The limitations of this study include the retrospective analysis from a single institution with incomplete collection of PROMIS-10 and EORTC QLQ-H&N35 data. The strengths include uniform oral cavity OAR avoidance structure contouring, prospective collection of CTCAE v4.03 adverse events, and completeness of the CTCAE v4.03, opioid use, and hospitalization data.

In RT treatment planning for head and neck cancer, the principle of “as low as reasonably achievable” should be followed when planning for doses to organs at risk such as the parotid glands, oral cavity, submandibular glands, floor of mouth, and oral tongue in order to minimize acute and late dysgeusia, dysphagia, xerostomia, weight loss, use of opioid pain medications, and hospitalization using current guidelines until there is additional information available regarding dose constraints for the oral cavity OAR avoidance structure [59]. A uniform consensus definition for the oral cavity OAR would be beneficial for future studies.

## 5. Conclusions

The oral cavity OAR avoidance structure as defined in this study predicts patients requiring opioid pain medication and hospitalization during and within 30 days of completing a course of RT for tumors of the head and neck region.

## Figures and Tables

**Figure 1 cancers-16-00349-f001:**
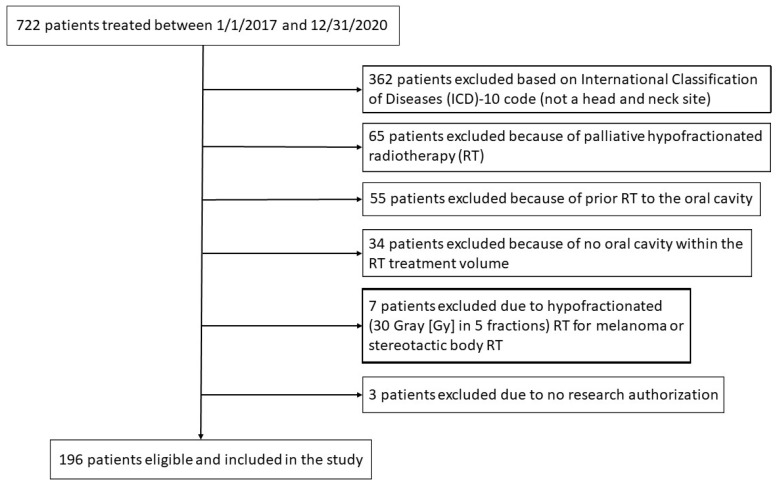
Consolidated Standards of Reporting Trials (CONSORT) diagram of patient inclusion and exclusion.

**Figure 2 cancers-16-00349-f002:**
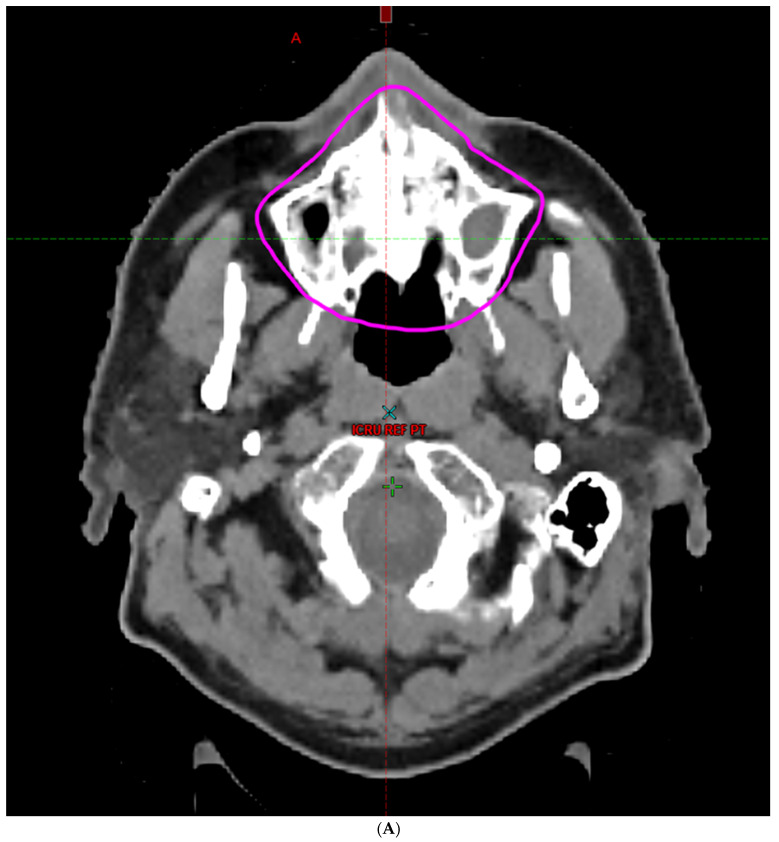
(**A**) Oral cavity organ-at-risk (OAR) avoidance structure definition (magenta line). Axial image at the level of the hard palate. The contouring of the OAR begins superiorly at the first sign of mucosa on the hard palate. (**B**) Oral cavity organ-at-risk (OAR) avoidance structure definition (magenta line). Axial image at the level of the alveolar ridges of the hard palate. The OAR includes the mucosa of the upper lip, gingiva of the maxillary alveolar ridge, hard palate, soft palate, and teeth. (**C**) Oral cavity organ-at-risk (OAR) avoidance structure definition (magenta line). Axial image at the level near the oral commissure. The OAR includes the mucosa of the upper lip, hard palate, gingiva, buccal mucosa, retromolar trigone, oral tongue, soft palate, and teeth. (**D**) Oral cavity organ-at-risk (OAR) avoidance structure definition (magenta line). Axial image at the level of the lower lip. The OAR includes the mucosa of the lower lip, buccal mucosa, gingiva, oral tongue, floor of mouth, retromolar trigone, soft palate, sublingual glands, and teeth. (**E**) Oral cavity organ-at-risk (OAR) avoidance structure definition (magenta line). Axial image at the level of the mandible. The OAR includes the mucosa of the lower lip, gingiva of the mandible, buccal mucosa, retromolar trigone, oral tongue, floor of mouth, and base of tongue, and the sublingual glands and teeth. (**F**) Oral cavity organ-at-risk (OAR) avoidance structure definition (magenta line). Axial image at the level of the inferior mandible and vallecula. The OAR includes the mucosa of the gingiva of the mandible, base of tongue, and vallecula. (**G**) Oral cavity organ-at-risk (OAR) avoidance structure definition (magenta line). Coronal image illustrating inclusion of the mucosa of the hard palate, alveolar ridges of the maxilla, alveolar ridges of the mandible, gingiva, buccal mucosa, floor of mouth and oral tongue, and the sublingual glands. (**H**) Oral cavity organ-at-risk (OAR) avoidance structure definition (magenta line). Sagittal image illustrating inclusion of the mucosa of the hard palate, soft palate, gingiva, lips, oral tongue, floor of mouth, and base of tongue and teeth.

**Table 1 cancers-16-00349-t001:** Patient, tumor, and treatment characteristics (*n* = 196).

**Characteristic**	
Male/female, *n*, (%)	142 (72.4)/54 (27.6)
Median age in years (IQR)	63.0 (55.1, 72.5)
Diabetes mellitus, *n*, (%)	15 (7.7)
Hypertension, *n*, (%)	83 (42.3)
Smoking status (none, former, current)	99, 87, 9
Ever smoker (no, yes)	99, 96
Leukoplakia	2
**Site**	
Oropharynx	53
Nasal cavity/paranasal sinus	27
Skin	27
Oral cavity	25
Major salivary gland	19
Larynx	12
Thyroid gland	12
Nasopharynx	8
Lacrimal gland/sac	3
Unknown primary (cervical lymph node metastasis)	3
Bone	2
Vascular (neck)	2
Conjunctiva	1
Hypopharynx	1
Trachea	1
**Histology**	
Squamous cell carcinoma	124
Adenoid cystic carcinoma	11
Adenocarcinoma	11
Carcinoma, NOS	9
Anaplastic carcinoma	7
Esthesioneuroblastoma	6
Melanoma	4
Merkel cell carcinoma	4
Unknown	4
Acinic cell carcinoma	3
Basal cell carcinoma	3
Epithelial–myoepithelial carcinoma	2
Paraganglioma	2
Undifferentiated carcinoma	2
Ameloblastoma	1
Angiosarcoma	1
Ewing sarcoma	1
Sinonasal undifferentiated carcinoma	1
**American Joint Committee on Cancer (AJCC) stage (7th and 8th edition) ***	
No AJCC staging system for site	22
Unknown	7
0	2
I	34
IA	1
IB	1
II	30
III	26
IIIA	3
IV	1
IVA	42
IVB	24
IVC	3
**Treatment**	
Glossectomy	22
Postoperative radiotherapy (RT)	110
Primary RT	86
Dose delivered, gray (Gy) ^†^, median (IQR)	60 (30, 80.4)
Dose per fraction, Gy, median (IQR)	2.0 (1.2, 2.25)
Total number of fractions, median (IQR)	30 (15, 35)
Protons	134
Photons	62
Cytotoxic chemotherapy, *n*, (%)	89 (45.4)
Mean oral cavity organ-at-risk (OAR) dose, Gy, median (IQR)	19.9 (9.65, 30.57)
Mean oral cavity OAR volume, cubic centimeters (cc), median (IQR)	309 (264, 356)
Mean total parotid gland dose, Gy median (IQR)	21.01 (5.89, 31.05)
Mean right parotid gland dose, Gy, median (IQR)	19.45 (3.38, 29.45)
Mean left parotid gland dose, Gy, median (IQR)	19.57 (4.65, 33.85)
Mean right submandibular gland dose, Gy, median (IQR)	39.98 (0.48, 61.14)
Mean left submandibular gland dose, Gy, median (IQR)	44.08 (0.28, 61.13)

* Depending on the year of diagnosis. ^†^ Gy Relative Biological Effectiveness (RBE) (1.1).

**Table 2 cancers-16-00349-t002:** Correlation between maximum dose (D_max_), mean oral cavity OAR dose, and V10Gy–V70Gy.

Pearson Correlations *	V10Gy	V20Gy	V30Gy	V40Gy	V50Gy	V60Gy	V70Gy
D_max_	0.33243	0.42786	0.46508	0.47391	0.43368	0.37524	0.30862
Mean	0.82351	0.90894	0.92801	0.89423	0.86955	0.82091	0.46076

* All reported correlations are statistically significant.

**Table 3 cancers-16-00349-t003:** Multivariable modeling—variables associated with toxicity.

Toxicity	Time Point	Variable	OR (95% CI)	*p*-Value	C-Statistic
CTCAE v4.03 dehydration > grade 1	End of treatment	Cytotoxic chemotherapy	5 (3–12)	<0.001	0.77
		Mean total parotid dose	1 (1–2)	0.044	0.77
CTCAE v4.03 dry mouth > grade 1	End of treatment	Mean right SMG dose	1 (1–1)	<0.001	0.68
CTCAE v4.03 dysgeusia > grade 1	End of treatment	Mean right SMG dose	1 (1–2)	<0.001	0.71
CTCAE v4.03 dysphagia > grade 1	End of treatment	Mean left SMG dose	1 (1–2)	0.029	0.84
		Mean right SMG dose	1 (1–2)	0.006	0.84
		Protons	0.3 (0.1–0.6)	0.003	0.84
	3 months	Mean left SMG dose	1 (1–2)	0.001	0.76
		Protons	0.4 (0.1–0.9)	0.029	0.76
CTCAE v4.03 salivary duct inflammation > grade 1	End of treatment	Mean left SMG dose	2 (1–2)	0.002	0.87
		Mean right SMG dose	2 (1–2)	0.002	0.87
		Protons	0.4 (0.2–1)	0.047	0.87
Weight loss ≥ 10%	3 months	Mean total parotid dose	2 (1–2)	<0.001	0.71
	6 months	Mean total parotid dose	2 (1–3)	0.002	0.79
		Mean right SMG dose	1 (1–2)	0.037	0.79
	1 year	Mean total parotid dose	2 (1–3)	0.001	0.79
		Mean right SMG dose	1 (1–2)	0.04	0.8
	2 years	Mean left SMG dose	1 (1–2)	0.005	0.7
EORTC saliva scale decrease ≥ 10 pts	1 year	Mean right SMG dose	1 (1–1)	0.028	0.67
	2 years	Mean right SMG dose	1 (1–2)	0.032	0.71
Start of opioid pain medication	During or within 30 days of completing RT	Cytotoxic chemotherapy	4 (2, 11)	0.002	0.75
		Oral cavity OAR mean dose	2 (1–2)	0.008	0.75
Hospitalization	During or within 30 days of completing RT	Cytotoxic chemotherapy	2 (1–5)	0.022	0.68
		Oral cavity OAR mean dose	1 (1–2)	0.036	0.68

OR: Odds Ratio. CI: Confidence Interval. CTCAE: Common Terminology Criteria for Adverse Events. Mean: mean dose in gray (Gy) Relative Biological Effectiveness (RBE) (1.1) to the total parotid glands, right submandibular gland, or left submandibular gland. SMG: submandibular gland. RT: radiotherapy. OAR: organ at risk.

**Table 4 cancers-16-00349-t004:** Radiotherapy (RT) modality and oral cavity organ-at-risk (OAR) dose–volume histogram (DVH) statistics.

DVH Statistic	Photon (*n* = 62)	Proton (*n* = 134)	Total (*n* = 196)	*p*-Value ^1^
**D_max_ (cGy)**				0.035
Mean (SD)	5853 (1595)	6317 (1342)	6170 (1439)	
Median	6384	6399	6392	
Q1, Q3	5864, 6782	6255, 7136	6208, 6991	
Range	(178–7526)	(0.1–8543)	(0.1–8543)	
**Mean (cGy)**				<0.001
Mean (SD)	2878 (1565)	1805 (1198)	2145 (1412)	
Median	2767	1801	1990	
Q1, Q3	1793, 4108	830, 2515	965, 3057	
Range	(31–5915)	(0.0–5271)	(0.0–5915)	
**V1000cGy**				<0.001
Mean (SD)	238 (106)	136 (83)	168 (102)	
Median	251	124	160	
Q1, Q3	175, 315	74, 207	87, 239	
Range	(0–401)	(0–335)	(0–401)	
**V2000cGy**				<0.001
Mean (SD)	178 (110)	104 (71)	128 (92)	
Median	176	93	117	
Q1, Q3	90, 267	47, 159	59, 183	
Range	(0–381)	(0–302)	(0–381)	
**V3000cGy**				<0.001
Mean (SD)	127 (103)	83 (61)	97 (80)	
Median	101	71	81	
Q1, Q3	34, 210	35, 127	35, 143	
Range	(0–368)	(0–270)	(0–368)	
**V4000cGy**				0.028
Mean (SD)	90 (98)	66 (53)	73 (71)	
Median	55	55	55	
Q1, Q3	5, 155	26, 103	15, 109	
Range	(0–345)	(0–247)	(0–345)	
**V5000cGy**				0.037
Mean (SD)	69 (87)	49 (46)	55 (62)	
Median	17	40	38	
Q1, Q3	1, 126	16, 77	5, 87	
Range	(0–310)	(0–225)	(0–310)	
**V6000cGy**				0.004
Mean (SD)	49 (70)	27 (36)	34 (50)	
Median	4	12	10	
Q1, Q3	0, 78	2, 37	0.4, 48	
Range	(0–255)	(0–200)	(0–255)	
**V7000cGy**				0.872
Mean (SD)	7 (26)	7 (20)	7 (22)	
Median	0	0	0	
Q1, Q3	0, 0	0, 0.1	0, 0	
Range	(0–158)	(0–159)	(0–159)	

^1^ two-sample *t*-test. Dmax: maximum dose. cGy: centigray. SD: standard deviation.

**Table 5 cancers-16-00349-t005:** Mean oral cavity OAR dose cut points and prediction of opioid pain medication use during and within 30 days of completing radiotherapy.

Mean Oral Cavity OAR Dose Cut Point (cGy)	Sensitivity	Specificity	Positive Predictive Value	Negative Predictive Value	Accuracy
<460, ≥460	91% (72/79)	16% (7/43)	67% (72/108)	50% (7/14)	65% (79/122)
<1050, ≥1050	81% (64/79)	40% (17/43)	71% (64/90)	53% (17/32)	66% (81/122)
<1610, ≥1610	71% (56/79)	54% (27/43)	78% (56/72)	54% (27/50)	68% (83/122)

OAR: organ at risk. cGy: centigray.

**Table 6 cancers-16-00349-t006:** Mean oral cavity OAR dose cut points and predication of hospitalization during and within 30 days of completing radiotherapy.

Mean Oral Cavity OAR Dose Cut Point (cGy)	Sensitivity	Specificity	Positive Predictive Value	Negative Predictive Value	Accuracy
<685, ≥685	90% (36/40)	19% (29/156)	22% (36/163)	88% (29/33)	33% (65/196)
<1450, ≥1450	80% (32/40)	38% (60/156)	25% (32/128)	88% (60/68)	47% (92/196)
<1960 ≥1960	70% (28/40)	53% (83/156)	28% (28/101)	87% (83/95)	57% (111/196)

OAR: organ at risk. cGy: centigray.

## Data Availability

The data presented in this study are available on request from the corresponding author. The data are not publicly available due to privacy restrictions.

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
