# Peer review of "Mean Oral Cavity Organ-at-Risk Dose Predicts Opioid Use and Hospitalization during Radiotherapy for Patients with Head and Neck Tumors"

_cancers, 2024, doi:10.3390/cancers16020349_

Round 1

Reviewer 1 Report

Comments and Suggestions for Authors

This paper describes associations between RT dose to a defined oral cavity organ at risk avoidance structure, provider and patient reported outcomes, opioid use, and hospitalization. Multivariable modeling revealed the mean dose to the oral cavity organ risk was significantly associated with opioid use and hospitalization. This study is well done with a relatively high N for the question with appropriate statistical methods. The manuscript has significant findings that are potentially clinically impactful. I expect this manuscript to be well cited given the rigor of the analysis. 

Author Response

Thank you for your positive and kind review of our work.

Reviewer 2 Report

Comments and Suggestions for Authors

The paper “Mean Oral Cavity Organ at Risk Dose (OAR) Predicts Opioid Use and Hospitalization During Radiotherapy for Patients with Head and Neck (H&N) Tumors by Robert L. Foote et al. is quite interesting in that it reports a retrospective study on the evaluation of the association between radiotherapy dose to a defined oral cavity organ at risk avoidance structure and prospectively obtained, validated provider and patient reported outcomes, opioid use, and hospitalization in 196 patients treated with radiotherapy for tumors in the H&N region. To deliver treasured information for radiotherapy treatment planning (TP) for the reduction in adverse events, the authors identified a statistically significant association between radiotherapy dose to the oral cavity organ at risk structure, opioid use, and hospitalization. The results of this study may be beneficial in RT TP for patients with tumors of the H&N region to reduce the need for opioid use and hospitalization during treatment.

The work is of a good standard and I would recommend cancers to publish this study.

I have only few comments, or suggestions as follows:

1. Please revise the figure caption and label the region name on images of Figure 2. Figure 2 shows the Oral Cavity Organ at Risk (OAR) Avoidance Structure. There are eight medical images in Figure 2 and each image can be the subfigure ((1)~(8)) of it. I would suggest the authors to descript the subfigures in the content correspondingly. It will help the readers easily to figure out the OAR by this figure.

2. The following captures are some minor editing or missing citation errors, please see attached file and revise them.

Comments on the Quality of English Language

No comments.

Author Response

Thank you for your thoughtful and thorough review of our manuscript.

As you have suggested, Figure 2 has now been revised as Figure 2A, 2B, 2C, 2D, 2E, 2F, 2G, and 2H.  Each region has been labeled and named and a description added.

I am unable to find the captures/attached file containing the minor editing and missing citation errors.  Please submit again and I will be happy to address those too.

Reviewer 3 Report

Comments and Suggestions for Authors

1. This is a retrospective study using existed medical record to analyze the association between mean radiation dose to oral cavity and opioid use/hospitalization during RT for head and neck cancer patients.

2. This study showed significant correlation between mean oral cavity dose and opioid use/hospitalization. Although not surprising, the study with a adequate patient number, still provides some evidence in predicting the severity of RT acute toxicity.

3. This study added some new evidence to show the association.

4. In the discussion session, the authors should make some comparison between the current study to the previous studies on the topic. Since the major finding is the association between opioid use and hospitalization, the previous studies on grading severity could be compared.

5. The conclusions are consistent with the evidence and arguments presented and they have addressed the main question posed.

6. The references are appropriate.

7. Tracheal cancer(ICD 10: C33) was included in the study. How could the authors defined which tracheal cancer to be included? Since most of the trachea is located in the thoracic cage. Similar issue existed in skin and bone cancer.

8. Please explain why there was only 1 hypopharyngeal cancer included in the study. This proportion is really bizzare.

Author Response

Thank you for your thoughtful and thorough review of our manuscript.

In the discussion section, we have added comparisons between our current study and the previous studies as you suggested.  This added discussion is highlighted in the revised manuscript.

With regard to your question about which tracheal, skin, and bone cancers were included in the study, as stated in the Materials and Methods section, inclusion criteria required the oral cavity to be within the treatment volume, and exclusion criteria included no oral cavity within the treatment volume.  Patients with tracheal cancer, skin cancer or bone cancer in which the extent of the primary tumor or lymph node involvement required the oral cavity to be within the treatment volume were included in the study.  There was just one tracheal cancer in the study and two bone (mandible) cancers.  It is not unusual for advanced facial skin cancers with regional lymph node metastases to have some of the oral cavity included in the treatment volume.  There was only one hypopharynx cancer included in the study because the extent of the treatment volume included the oral cavity.  For the majority of hypopharynx cancers, the oral cavity is not included within the treatment volume.

"Inclusion criteria included all patients with benign or malignant tumors within the head and neck region receiving RT to the oral cavity by one of the authors (RLF) between 1/1/2017 and 12/31/2020."

"Exclusion criteria included a palliative course of hypofractionated RT, prior RT to the oral cavity, no oral cavity OAR within the RT treatment volume (no dose to the oral cavity), hypofractionated RT for melanoma, hypofractionated stereotactic body RT, and no consent provided to use medical records for research purposes as provided either by Department of Radiation Oncology Registry (IRB #15-000136, March 15, 2015) or by Minnesota state statute."  

Reviewer 4 Report

Comments and Suggestions for Authors

These authors conducted a retrospective study to evaluate the association between radiotherapy dose to a defined oral cavity organ at risk (OAR) avoidance structure and once prospectively obtained, to examine patient reported outcomes, opioid use, and hospitalization in patients treated with radiotherapy for head and neck tumors. This study is novel and adds to the literature on the side effects of radiotherapy in treated head and neck cancer patients. This study also adds important information for RT dose planning. The writing is clear, organized, and concise. The Background/lit review is adequate and appropriate. Methodology is clear. Results are clear, including figs and tables. Discussion is thoughtful and includes relevant literature. Limitations are noted and appropriate. Just one comment:

1    References #31 and #32 are listed twice.

Author Response

Thank you for your positive and kind review.  Thank you for noticing the duplication of references #31 and #32.  The duplicate references have been removed.